# Colon Organoids as Experimental Models to Study the Effect of Micro-Nanoparticles as a Driver of Early-Onset Colon Cancer

**DOI:** 10.3390/cells15010040

**Published:** 2025-12-25

**Authors:** Zahra Heydari, Gobinda Sarkar, Lauren Helgeson, Estela Mariel Cruz Garcia, Alexandra Ros, Khashayarsha Khazaie, Lisa Boardman

**Affiliations:** 1Department of Gastroenterology and Hepatology, Mayo Clinic, Rochester, MN 55905, USA; sarkar.gobinda@mayo.edu (G.S.); helgeson.lauren@mayo.edu (L.H.); 2School of Medicine, University of Puerto Rico, San Juan, PR 00921, USA; estela.cruz1@upr.edu; 3The Biodesign Institute, Arizona State University, Tempe, AZ 85281, USA; alexandra.ros@asu.edu; 4School of Molecular Sciences, Arizona State University, Tempe, AZ 85287, USA; 5Department of Immunology, Mayo Clinic, Scottsdale, AZ 85259, USA; khazaie@mayo.edu

**Keywords:** organoid, colorectal cancer, micro- and nanoplastics, early-onset CRC

## Abstract

Early-onset colorectal cancer (EOCRC) in people < 50 years of age has been rising globally, yet its causes remain unknown. Emerging evidence suggests that environmental factors, including exposure to micro-and nanoplastics (MNPs), may contribute to colorectal carcinogenesis. MNPs can enter the gastrointestinal tract through ingestion, translocate across the epithelial barrier via endocytosis or paracellular pathways, and interact directly with epithelial and immune cells. Once internalized, they may generate events associated with tumor initiation including oxidative stress, disruption of membrane integrity, pro-inflammatory signaling, and disruption of genomic and epigenomic stability. Patient-derived colorectal organoids offer a physiologically relevant and scalable 3D model that closely mimics the cellular architecture and genetic landscape of primary tumors. We highlight how organoid models can be leveraged to study the impact of MNPs on the key processes of inflammation, DNA damage, senescence, and epigenetic modifications. Furthermore, we discuss the application of organoid-based systems to model EOCRC driven by environmental exposures, including the integration of organoid platforms with high-throughput assays, omics profiling, and microfluidics to better capture MNP-induced pathogenic mechanisms. Altogether, colorectal organoids provide a powerful bridge between environmental plastic exposure and EOCRC etiology, offering a tractable platform to identify mechanistic pathways and potential biomarkers of early disease.

## 1. Introduction

Colorectal cancer (CRC) remains the third most diagnosed cancer globally, with an estimated 1.9 million new cases reported in 2020. While traditionally considered a disease of older adults, an alarming epidemiological shift has been observed: the incidence of early-onset colorectal cancer (EOCRC), defined as CRC occurring in individuals under 50 years of age, is steadily rising worldwide [1,2]. Notably, this increase is most pronounced in left-sided tumors, particularly affecting the rectum, rectosigmoid, and sigmoid colon [3]. Projections suggest that within the next decade, up to one-quarter of rectal cancer cases may occur in individuals under 50 years old [4]. Although the precise etiology of EOCRC remains uncertain, the consistent rise across generations, especially among those born after 1960, suggests that environmental and lifestyle factors may be key contributors [5,6]. Hypothesized influences include rising rates of obesity, sedentary behavior, Western dietary patterns, early-life antibiotic exposure, and hormonal changes associated with widespread contraceptive use [7,8]. Attention is also being directed toward the role of the gut microbiota and its interactions with the intestinal epithelium, particularly the integrity of the colonic mucus barrier. This protective layer not only separates luminal bacteria from the host epithelium but also serves as a niche for microbial communities, some of which may exert pro-inflammatory or carcinogenic effects. Disruptions to this delicate mucosal environment are increasingly implicated in the pathogenesis of sporadic CRC, including EOCRC, underscoring the need for deeper mechanistic insights into environmental and microbiome-related risk factors [9,10,11].

The extensive production and inadequate disposal of plastic materials have led to the widespread environmental presence of micro- and nanoplastics (MNPs), raising growing concerns about their potential effects on human health. Although various recycling strategies have been introduced globally, only a small fraction of plastic waste, approximately 9%, is effectively recycled, with the remainder either incinerated or released into the environment [12]. Consequently, mechanical, thermal, and photo-oxidative degradation of plastic waste continues to generate MNPs, which have now been detected across diverse environmental matrices, including soil, surface and groundwater, and even within the human food supply [13,14]. Recent studies have identified MNPs in common food items such as seafood, grains, meat, and dairy, indicating that chronic human ingestion is not only ongoing but likely to intensify [13,14]. While the long-term health implications remain under investigation, accumulating evidence from in vitro and in vivo models suggests that MNPs may disrupt gut homeostasis through mechanisms involving oxidative stress, inflammation, apoptosis, and impaired epithelial barrier function. This persistent exposure promotes oxidative stress through elevated reactive oxygen species (ROS), triggering inflammation, cellular senescence, and pathways linked to intestinal disorders and carcinogenesis. Inflammatory responses further exacerbate cellular dysfunction, potentially contributing to cancer development [15,16]. Emerging evidence from in vitro and in vivo studies suggests MNPs disrupt gut epithelial homeostasis via oxidative and inflammatory mechanisms, with bioaccumulation observed across multiple organs [17,18].

CRC organoids provide a physiologically relevant model of the human gut, capturing key features such as cellular diversity, tissue architecture, and epithelial polarity [19]. These systems allow precise investigation of environmental factors, including MNPs, impact gut homeostasis, barrier function, and early carcinogenic processes. By linking controlled mechanistic studies to clinically relevant outcomes, organoid models offer a powerful tool to explore the role of MNPs in EOCRC. Integrating organoid-based approaches into research thus enables a deeper understanding of environmental contributions to colorectal carcinogenesis [20,21,22].

In the following sections, we explore how MNPs interact with the gastrointestinal environment and influence processes relevant to colorectal carcinogenesis, with particular attention to their potential role in EOCRC. We highlight emerging insights into how these particles are internalized by intestinal cells, how they may contribute to epithelial dysfunction and malignant transformation, and how advanced organoid-based models are being used to capture these effects in physiologically relevant systems. We further consider how microplastic exposure intersects with broader biological processes such as aging and telomere dynamics, aiming to provide an integrative perspective that connects environmental exposures with molecular and clinical dimensions of EOCRC.

## 2. Types of Micro- and Nanoplastics, Biological Relevance and Mechanisms of Uptake

MNPs are heterogeneous particles varying widely in size, shape, polymer composition, and surface chemistry, which collectively influence their environmental persistence, biological interactions, and toxicological outcomes. Common polymer types include polyethylene (PE), polypropylene (PP), polystyrene (PS), polyvinyl chloride (PVC), and polyethylene terephthalate (PET), each with distinct physicochemical properties affecting their degradation and cellular uptake [23,24,25]. Humans are primarily exposed to MNPs through ingestion, inhalation, and, to a lesser extent, dermal contact. Additional exposure routes—including dermal contact (e.g., from textiles), as well as mucosal surfaces such as the eyes, ear passages, and urogenital tract, may also contribute, although their relative importance and dose relevance remain less well quantified. These routes are likely less central to colorectal exposure but warrant acknowledgment [24,26,27]. Once bioavailable, these particles can translocate to internal tissues, with their biological effects influenced by size, shape, composition, dose, duration of exposure, and the release of chemical additives. Among gastrointestinal sites, the colon is of particular concern due to prolonged particle retention, its dense and metabolically active microbiota, and emerging evidence linking chronic inflammation to colorectal carcinogenesis [28,29].

Nanoplastics (NPs), defined as plastic particles smaller than 100 nm, exhibit higher surface area-to-volume ratios and greater surface reactivity than microplastics (MPs; ~100 nm–5 mm), resulting in enhanced cellular penetration and bioactivity, whereas MPs tend to accumulate on tissue surfaces, causing mechanical irritation and inflammation with comparatively lower cellular uptake. Microplastics tend to accumulate on tissue surfaces, causing mechanical irritation and inflammation, but generally show lower cellular uptake due to their larger size [30]. MNPs interact with the intestinal epithelium through surface adhesion and cellular uptake, processes that can promote barrier dysfunction and local inflammation, thereby contributing to carcinogenic pathways [31,32]. In a recent study, Cheng and colleagues showed that NPs (~100 nm) are taken up via endocytosis and induce ferroptosis, a regulated cell death involving oxidative stress, lipid peroxidation, and glutathione depletion, mediated by the Fosl1-p53-Slc7a11 pathway [33]. Blocking this pathway mitigates ferroptotic cell death, underscoring its central role in mediating NPs toxicity. In contrast, larger MPs (~10 µm) cause mechanical stress that activates the YAP pathway, leading to cytoskeletal changes and metabolic reprogramming from oxidative phosphorylation to anaerobic glycolysis, which is associated with inflammation [33,34].

Size-dependent intestinal effects of MNPs have also been demonstrated in vivo. In a controlled exposure study, C57BL/6J mice were administered polystyrene microplastics (PS-MPs) of 0.2, 1, or 5 μm daily (1 mg/kg body weight) for 28 days [35]. All sizes triggered oxidative stress, inflammatory cell infiltration, increased colonic permeability, reduced mucus secretion, and downregulation of tight-junction proteins (ZO-1, occludin, claudin-1), but the largest particles (5 μm) caused the most severe epithelial barrier disruption. In Caco-2 cells, 5 μm PS-MPs activated the ROS–NF-κB–NLRP3 inflammasome–MLCK signaling axis, leading to tight-junction disassembly. Antioxidant (NAC), NLRP3 inhibitor (MCC950), or MLCK inhibitor (ML-7) treatment attenuated these effects, confirming the involvement of ROS-dependent NF-κB/NLRP3/IL-1β/MLCK signaling in barrier injury [35].

Upon entering the human body, MNPs interact with target cells in ways that depend on particle size, surface chemistry, and the biological molecules they adsorb, such as proteins forming a “protein corona” that alters their cellular recognition and toxicity [36,37]. NPs tend to accumulate more readily and translocate efficiently, especially in the gut. Cellular uptake occurs mainly through active, energy-dependent endocytotic pathways, including clathrin- and caveolae-mediated endocytosis, macropinocytosis, and phagocytosis, which vary by particle size, surface charge, and cell type. Positively charged particles have been reported to have higher uptake and cytotoxicity due to enhanced binding to negatively charged cell surfaces, while negatively charged particles face uptake resistance [38]. After internalization, MNPs often accumulate in endosomal and lysosomal compartments, potentially releasing toxicants intracellularly. Importantly, different MNP types and surface modifications lead to varied uptake mechanisms and toxicological profiles across cell types, emphasizing that size, chemistry, and environmental transformations critically influence their biological effects, particularly in colorectal tissues, where diverse MNP exposure may drive inflammation, barrier disruption, and carcinogenesis (Figure 1) [39].

In vitro studies using human colon organoids and small intestine epithelial cell lines have confirmed that NPs can accumulate intracellularly, even localizing to the cytoplasm and nucleus under certain conditions. Two biological modifiers strongly influence these interactions. First, the mucus layer is a critical biophysical barrier: native human mucus substantially reduces microplastic translocation and cellular toxicity, whereas surface functionalization or protein coating (for example, streptavidin modification) can enhance mucus penetration and increase particle–cell contact, inflammation and cytotoxicity [40]. Second, gastrointestinal digestion alters particle surfaces by forming a digestion-associated protein corona. This corona, enriched in coagulation factors, apolipoproteins, and vitronectin, persists in serum-containing media and markedly enhances uptake of particles < 500 nm by THP-1-derived macrophages by 4–6 times. Such biotransformation highlights how digestion can facilitate recognition and internalization by both epithelial and immune cells [31]. Together, these mechanistic insights emphasize the importance of particle physiochemistry, host barriers, and corona formation in determining MNP fate in the gut, factors that warrant focused study in the context of chronic exposure and early-onset colorectal cancer.

## 3. Early-Onset Colorectal Cancer: Clinical Significance and Molecular Signature

EOCRC, defined as CRC diagnosed before the age of 50, has been steadily increasing over recent decades and now accounts for about 10% of all CRC cases annually, with a concurrent rise in mortality (Table 1). In contrast, average-onset colorectal cancer (AOCRC), defined as CRC diagnosed at ≥50 years, has shown stable or declining incidence in many regions, a trend largely attributed to the impact of population screening programs [41,42]. While the pathogenesis of EOCRC is well understood in hereditary cases, approximately 80% of EOCRC patients do not carry known genetic mutations linked to CRC predisposition, and therefore are classified as sporadic tumors [43,44]. These sporadic EOCRC cases are often more aggressive, with a higher likelihood of early metastasis and poorer prognosis. Historically, these were frequently diagnosed at advanced stages because screening programs had generally excluded people under 50 unless they had a strong family history of colorectal cancer. The European Society of Gastrointestinal Endoscopy (ESGE) guidelines were one example of this. However, as the rise in EOCRC has become recognized, these guidelines are evolving to meet this need. The United States Preventive Services Task Force (USPSTF) updated their screening recommendations in May of 2021 to offer average risk screening to begin at age 45, thus adopting the American Cancer Society guideline recommending this change since 2018 which ultimately provided the endorsement needed by the USPSTF for insurance coverage for average risk CRC screening for younger individuals [45]. This highlights the urgent need for new biomarkers and prevention strategies that can identify average-risk individuals in EOCRC screening to enable earlier detection and improved outcomes [46,47]. In the U.S., EOCRC incidence is growing faster among younger cohorts (20–44 years) than among those aged 45–54 (average annual percentage changes (AAPC): 1.51 vs. 0.73), with increases particularly marked in proximal colon tumors and in late-stage diagnoses, which portend worse prognosis. Alarmingly, mortality rates in the 20–44 age group are also rising significantly (AAPC ≈ 0.93 vs. stable in older groups) [48].

This epidemiologic trend parallels the explosion of global plastic pollution. Researchers hypothesize that MNPs may disrupt the intestinal environment and contribute to EOCRC risk. MNPs have been implicated in mucus layer degradation, barrier dysfunction, microbiome dysbiosis, oxidative stress, and DNA damage, all recognized drivers of colorectal carcinogenesis [52]. Although direct causal evidence remains emergent, the time course of plastic dissemination aligns with rising EOCRC rates, suggesting environmental exposure as a plausible contributor [53].

The molecular landscape of EOCRC reveals notable distinctions from late-onset CRC, suggesting unique pathogenic pathways. While some overlap exists, younger patients more frequently present with microsatellite instability (MSI-H) and deficient MMR (dMMR) phenotypes, particularly in hereditary cases, although the majority of sporadic EOCRC are microsatellite stable (MSS) [54]. Sporadic EOCRC often shows chromosomal instability (CIN) and harbors somatic alterations in canonical drivers such as *APC*, *KRAS*, and *TP53*, but at differing frequencies than in late-onset CRC. For example, *APC* mutations are less common in EOCRC, potentially leading to alternative routes of WNT pathway dysregulation, while *TP53* mutations occur more frequently, particularly in left-sided tumors. EOCRC tumors also demonstrate distinct copy number variations, including gains in 8q (*MYC*), 20q (*SRC*, *AURKA*), and 13q, which may contribute to their aggressive behavior [54,55,56].

Epigenetically, EOCRC tends to exhibit LINE-1 hypomethylation, reflecting global genomic instability, while the CpG island methylator phenotype (CIMP) appears less frequent compared to late-onset MSI-high CRC [57]. Transcriptional analyses further suggest enrichment of proliferative and stem-like signatures, with alterations in pathways such as MAPK, PI3K/AKT, and JAK/STAT, which regulate survival and inflammatory responses [54,58]. Emerging data also implicate the gut microbiome, particularly exposure to colibactin-producing Escherichia coli, which leaves characteristic mutational imprints in EOCRC genomes, supporting a role for early-life environmental exposures in tumor initiation [59].

These molecular signatures collectively differentiate EOCRC from late-onset disease, highlighting both unique vulnerabilities (e.g., higher TP53 mutation burden, CIN-driven aggressiveness) and potential therapeutic opportunities, such as immunotherapy for MSI-H tumors and pathway-targeted interventions for CIN-driven cancers [60,61,62].

## 4. Microplastics and Colorectal Carcinogenesis: Integrating Telomere Biology and EOCRC Molecular Features

Emerging evidence suggests that MNPs may be relevant to biological processes implicated in EOCRC, whose rising incidence has been linked to environmental and lifestyle risk factors. EOCRC is distinguished by features such as increased chromosomal instability, high *TP53* mutation burden, and immune-microenvironment shifts, which overlap mechanistically with MNP-induced effects on barrier integrity, inflammation, and dysbiosis [54,63]. Given that younger populations may face disproportionately high plastic exposure through diet and consumer products, MNPs represent a plausible, understudied contributor to the upward EOCRC trend.

In a controlled murine study, chronic oral exposure to PS-MPs for 28 days resulted in marked downregulation of tight-junction proteins, including zonula occludens-1 (ZO-1), occludin, and claudin-1, accompanied by increased colonic permeability, elevated oxidative stress, and upregulation of pro-inflammatory cytokines interleukin (IL)-1β and IL-6. These effects were most pronounced with larger particles (5 μm) and were mediated via a reactive oxygen species (ROS)-dependent NF-κB/NLRP3/IL-1β/myosin light chain kinase (MLCK) signaling pathway [35] (Figure 2). Similarly, Xie et al. demonstrated that exposure to PS-MPs in mice and colon organoid cultures induced crypt hyperproliferation and mucosal overgrowth, with a reduction in goblet cell numbers and increased expression of stem cell-associated markers proliferating cell nuclear antigen (PCNA) and *c-Myc*, indicative of Notch pathway activation. This was accompanied by modest but significant increases in IL-1β and IL-6 expression, as well as exacerbation of dextran sulfate sodium–induced colitis, characterized by weight loss, mucosal ulceration, and histological evidence of epithelial injury. MNP-induced activation of the Notch signaling pathway may influence goblet cell differentiation, helping explain why mucus secretion can appear variable despite an overall reduction in goblet cell numbers [64]. While these studies do not directly model EOCRC, they highlight epithelial and inflammatory responses that may be relevant to early tumor-promoting conditions.

Ex vivo and in vitro models provide corroborating evidence of these barrier-disruptive and pro-inflammatory effects. Donkers et al. reported that exposure of human colon explants to 1–10 μm PS spheres and high-density polyethylene (HDPE) fragments significantly reduced tissue viability and barrier function, whereas nylon fibers specifically induced IL-6 release from colonic epithelial cells. Confocal microscopy confirmed the translocation of fluorescently labeled MPs across the epithelial barrier, supporting the concept of physical penetration and direct epithelial interaction [65]. Alterations in the gut microbiota represent another plausible mechanism by which MNP exposure may influence colorectal disease risk. Using a human colonic fermentation model, Tamargo et al. demonstrated that PET microplastics, following simulated gastrointestinal digestion, induced significant shifts in microbial community composition. These changes were characterized by alterations in the relative abundance of key bacterial taxa, with evidence suggesting that certain gut microorganisms can adhere to PET microplastics and form biofilms. Such biofilm formation may facilitate prolonged MNP–microbe interactions, modify metabolic outputs, and promote dysbiosis, conditions increasingly recognized as contributing to colorectal carcinogenesis through inflammation, genotoxin production, and epithelial barrier disruption [66]. Li et al. exposed mice to PE microplastics (6–600 μg/day) in drinking water for five weeks, resulting in dose-dependent gut dysbiosis characterized by increased *Staphylococcus* and reduced *Parabacteroides*. These shifts were accompanied by elevated serum IL-1α, altered Th17/Treg ratios, and colonic inflammation with upregulated *TLR4*, AP-1, and *IRF5* expression, indicating microbiota–immune axis disruption [67].

Telomeres, the repetitive DNA–protein complexes at chromosomal ends, act as guardians of genomic stability. With each cell division, telomeres progressively shorten due to incomplete replication, ultimately triggering replicative senescence when critically short. Telomere erosion is widely recognized as a hallmark of aging and a driver of genomic instability, a key mechanism underpinning carcinogenesis [68,69]. Importantly, telomere dynamics are tissue specific. A study using quantitative fluorescence in situ hybridization (Q-FISH) and quantitative PCR (Q-PCR) in normal colorectal tissues demonstrated that telomere length declines steadily with age until approximately 60–70 years. Intriguingly, beyond this age, an unexpected positive association between age and telomere length was observed exclusively in colonic epithelial cells, but not in stromal cells. This paradoxical finding suggests a selective survival effect, individuals who reach older ages may inherently possess longer epithelial telomeres, potentially conferring resilience against age-related genomic instability. Peripheral blood lymphocytes, however, displayed a consistent inverse correlation between telomere length and age, without a rebound in the elderly. This highlights the organ- and cell-type specificity of telomere biology and underscores the limitations of using leukocyte telomere length as a surrogate for tissue-specific telomere dynamics [68].

Recent studies suggest that accelerated telomere shortening may predispose individuals to EOCRC. A case–control study reported significantly shorter leukocyte telomere length in EOCRC patients (mean ~122 kb) compared to age-matched controls (~296 kb), supporting a link between premature biological aging and EOCRC susceptibility. Whole-exome sequencing further identified germline variants in several telomere maintenance genes, *hTERT*, *POT1*, *TERF2*, *TERF2IP*, significantly associated with EOCRC susceptibility. These findings suggest that shorter telomeres and genetic dysregulation of telomere-protective proteins may contribute to EOCRC development [46]. A large case-matched study (31,164 individuals) evaluated “accelerated aging” using the biomarker PhenoAge across both EOCRC and late-onset colorectal cancer (LOCRC). They found that for each year PhenoAge exceeded chronological age, the odds of EOCRC increased by 7%, while the increase in LOCRC risk was just 1%. This highlights that subclinical biological aging may be especially relevant in EOCRC, even if telomere length wasn’t directly measured in this study. Moreover, evidence remains inconclusive; a large prospective cohort study failed to detect significant systemic telomere differences between EOCRC patients and controls [70]. While tissue-specific telomere shortening in colonic epithelial cells is well-documented, studies investigating peripheral blood leukocyte telomere length (PBL-TL) show mixed results. A meta-analysis of seven studies (eight datasets) explored the association between PBL-TL and CRC risk: Prospective studies (4 datasets) found no significant association between PBL-TL and CRC risk (OR = 1.01, 95% CI: 0.77–1.34; I^2^ = 30%). Retrospective studies (4 datasets) indicated a nonsignificant trend toward increased CRC risk with shorter PBL-TL (OR = 1.65, 95% CI: 0.96–2.83; I^2^ = 96%). Female-only subgroup analyses also revealed no significant correlation (OR = 1.17, 95% CI: 0.72–1.91). These findings suggest that PBL-TL may not be a reliable biomarker for CRC or EOCRC detection. This discrepancy likely reflects tissue-specific telomere dynamics, where leukocyte telomere length may not accurately mirror telomere *TERF2* occurring in colonic epithelial or stem cells [71]. This suggests that systemic telomere measures may poorly reflect colonic epithelial aging, emphasizing the need for tissue-specific assessment and integration of models like organoids to better understand EOCRC pathophysiology.

Emerging evidence suggests that exposure to MNPs during pregnancy may influence biological aging-related processes, potentially through telomere disruption. Recent bibliometric mapping of research trends from 2010 to 2024 underscores oxidative stress as the central mechanism mediating microplastics-induced toxicity, revealing its pivotal role in linking inflammation, gut microbiota dysbiosis, and apoptotic pathways, while also highlighting a growing research focus on molecular pathways and antioxidant-based interventions [72,73]. A large cohort study of 1121 pregnant women from Shenyang, China, demonstrated that placental accumulation of polyvinyl chloride (PVC), polypropylene (PP), and polybutylene succinate (PBS), detected at a median of ~15 particles per 10 g tissue, was associated with significantly shortened telomeres in both umbilical cord blood (*β* = −0.13 and −0.14 for PVC and PBS, respectively; *p* ≤ 0.01) and placental tissue (*β* = −0.13 for PP, *p* < 0.001) [74].

Mechanistically, MNP exposure is known to trigger oxidative stress, inflammation, and cellular senescence across diverse systems. A recent narrative review summarizes how MNP uptake by immune and structural cells leads to ROS production, DNA and protein damage, and pro-inflammatory signaling pathways that intersect with aging-related processes [75]. Placental explant studies further confirm that PS MPs cause time-dependent cytotoxicity, elevate levels of mitochondrial and total superoxide anions and hydrogen peroxide, reduce antioxidant defenses, and induce oxidative damage (e.g., elevated malondialdehyde and carbonylated proteins) [73]. Taken together, these findings strongly suggest that prenatal MNP exposure compromises telomere integrity, potentially inducing a “pre-aged” cellular phenotype from birth. Since telomere shortening is a key driver of genomic instability and tumorigenesis, such early-life molecular perturbations may represent a biologically plausible, but as yet unproven, contributor to EOCRC susceptibility later in life.

## 5. Colorectal Organoid Models and MNPs: State of the Field, Limitations, and Promise

Human colorectal organoids provide an advanced experimental model of gastrointestinal physiology and disease, as they closely replicate the native colon epithelium, including its architecture, stem cell niches, and diverse cell lineages. Emerging evidence highlights colorectal organoids as a powerful platform to investigate the mechanisms of MNP uptake, intracellular trafficking, and toxicity, particularly in the context of EOCRC. In the current section, we outline the different types of colorectal cancer organoid cultures currently available (Table 2).

Apical-Out organoids: Direct Luminal Exposure Models. A major innovation in organoid-based toxicology is the development of apical-out organoids, in which the luminal (apical) surface is oriented outward toward the culture medium (Figure 3). This configuration enables direct exposure of the epithelium to suspended MNPs, closely mimicking oral ingestion and supporting dynamic, high-resolution functional analyses [76]. In a recent study, apical-out small-intestinal organoids were combined with fluorescence lifetime imaging microscopy (FLIM) to track nanoparticle interactions in real time. Using pristine PMMA and polystyrene nanoparticles (<200 nm), the authors demonstrated topology-dependent uptake, with apical-out models showing faster and more uniform internalization compared to conventional basal-out organoids. FLIM further enabled ‘lifetime barcoding,’ allowing discrimination of particle types and mapping of interaction sites within the tissue microenvironment. Functionally, both short—(24 h) and long-term (72 h) exposures disrupted mitochondrial membrane potential and increased CXCL-8 secretion, reflecting early epithelial stress in intact 3D tissue. Together, these findings establish apical-out colorectal organoids as a highly powerful platform for modeling luminal exposure scenarios relevant to EOCRC [77]. While apical-out culture has been widely applied to tumor-derived and small-intestinal organoids, it is also feasible in normal human colon organoids, with careful handling in suspension to preserve epithelial polarity and viability. CRC-derived organoids are generally more robust in this configuration, facilitating broader experimental applications [76,78].Organoid-Derived Epithelial Monolayers: High-Throughput Barrier Models. To complement 3D systems, organoid-derived epithelial monolayers have been developed by dissociating organoids and seeding single cells onto permeable supports, where they reconstitute a polarized epithelial sheet. These models retain the diverse cell populations of colorectal organoids while offering enhanced accessibility to both apical and basolateral compartments (Figure 3). Importantly, they allow for quantitative assessment of barrier integrity, particle transport, and transepithelial permeability, parameters difficult to measure in 3D organoids [79]. A study using monolayers has shown that NPs as small as 50 nm can cross the epithelial barrier through clathrin-mediated endocytosis, preferentially accumulating in secretory and absorptive cell subsets. Co-exposure experiments with chlorpromazine, a clathrin pathway inhibitor, significantly reduced MNP internalization, underscoring the active, energy-dependent mechanisms underlying nanoparticle uptake. These monolayers also facilitate high-throughput screening of particle size, shape, and surface chemistry, helping to delineate structure–toxicity relationships at the human intestinal barrier [80]. Recent investigations using human intestinal or colon organoids have begun to illuminate how MNPs influence epithelial integrity, cell-specific uptake, and inflammatory signaling. M cells are specialized epithelial cells located in the follicle-associated epithelium of the gut that facilitate the transcytosis of luminal particles and antigens to underlying immune cells, thereby playing a key role in mucosal immunity and pathogen sampling. Chen et al. utilized human intestinal organoid–derived epithelial monolayers (with and without M cells) and demonstrated that particle uptake increases with size, concentration, and exposure time, and that M-cell–containing monolayers show significantly greater transport of larger particles and inflammatory cytokine release [81].Basal-Out organoids: Modeling Basolateral Exposure and Systemic Risk. Traditional basal-out organoids, where the apical surface faces inward toward the luminal space and the basal surface interfaces directly with the culture medium, are commonly employed in studies modeling systemic or basolateral exposure routes. While this configuration precludes direct luminal access, it remains invaluable for examining basolateral uptake mechanisms, intracellular trafficking, and epithelial–mesenchymal signaling. Indeed, this baseline cultural orientation has been foundational in modeling epithelial homeostasis and dysfunction in organoid systems [82].

To access the apical epithelium, basolateral-out organoids are typically cultured in an ECM scaffold and then manipulated by: (i) microinjection into the lumen (ii) mechanical disruption to expose the apical surface, or (iii) dissociation and re-seeding onto inserts to form 2D monolayers [83,84,85]. Given the complexity of EOCRC and the need for models that closely mimic the native tumor microenvironment, apical-out organoids offer significant advantages over apical-in and monolayer models. Their ability to maintain epithelial barrier integrity, facilitate polarized secretion, and interact with the microenvironment makes them a superior choice for studying EOCRC. While apical-in organoids and monolayer models have their applications, they may not fully recapitulate the complexities of EOCRC as effectively as apical-out organoids [78]. A recent study using human intestinal organoids demonstrated that ~50 nm PS-NPs, at concentrations of 10 and 100 µg/mL, preferentially accumulated in specific epithelial cell types, with particularly high uptake in secretory cells. This accumulation triggered apoptosis and inflammatory responses, implicating PS-NPs in direct epithelial injury. Mechanistic experiments revealed that clathrin-mediated endocytosis is a key pathway for PS-NP internalization, as co-treatment with the inhibitor chlorpromazine significantly reduced nanoparticle uptake in secretory cells. These findings not only clarify the cellular mechanisms underlying PS-NP uptake and toxicity in human intestinal epithelium but also suggest endocytosis inhibition as a potential strategy to mitigate nanoplastic-induced epithelial damage [80]. Park et al. conducted a comparative analysis using iPSC-derived colon organoids and found that smaller microplastics (50 nm) caused greater reductions in organoid viability (>20%), decreased size, and robust upregulation of genes related to inflammation, apoptosis, immunity, metabolism, and extracellular matrix remodeling, effects that were more pronounced than those induced by 100 nm particles [86].

**Table 2 cells-15-00040-t002:** Comparison of 3D organoid platforms for modeling MNP exposure in EOCRC research [87,88,89,90].

Feature	3D Apical-In Organoids	3D Apical-Out Organoids	Organoid-Derived 2D Transwell Monolayers
Polarity/Topology	Apical surface faces lumen; basolateral side exposed to medium	Apical surface faces outward; direct access to luminal side	Flat monolayer; apical and basolateral compartments separated
MNP Exposure Route: [Note: Exposure efficiency varies with MNP size and charge]	Microinjection into lumen or basolateral exposure	Direct addition to apical surface in medium	Apical exposure via top chamber or basolateral via bottom chamber
Advantages	Preserves stem cell niche and crypt-villus architecture; high physiological relevance	Directly mimics luminal exposure; easier MNP dose control; suitable for mucus and microbiome co-culture	High throughput; Transepithelial/Transendothelial Electrical Resistance (TEER) measurements possible; suitable for permeability, transport and signaling studies
Limitations	Technically challenging microinjection; lower throughput; difficult imaging of apical responses	Loss of some niche signaling; fragile polarity (Polarity can revert over time; difficult to sustain beyond several passages); limited chronic exposure modeling	Loss of 3D architecture; reduced stem cell maintenance; limited crypt-villus physiology
Best Applications	Studying stem cell niche response, genomic instability, telomere dynamics	Modeling MNP luminal exposure, epithelial barrier disruption, immune activation	Transport studies, high-throughput MNP screening, co-culture with immune or stromal cells
Telomere Assays	qPCR, Q-FISH, and STELA possible; reflects crypt cell-specific effects	qPCR and Q-FISH feasible; easier spatial mapping of apical vs. basal shortening	qPCR-based telomere length analysis; suitable for time-course telomere attrition studies
Barrier Function Tests	Limited; TEER not possible	Moderate; measure dye permeability but no TEER	Direct TEER measurement; tracer assays (FITC-dextran, Lucifer Yellow), Integration with microfluidic chips possible for real-time monitoring (especially Transwell)
Co-Culture Potential	Limited but possible with immune cells in basolateral medium	Good compatibility with mucus-producing bacteria and immune cells without microinjection.	Easiest integration of immune, stromal, or endothelial co-cultures
Physiological Fidelity	Highest, retains crypt-villus gradient	Moderate, partially recapitulates lumenal exposure	Lowest, but controllable environment
Chronic Exposure Modeling	Feasible with passaging; better reflects cumulative aging	Possible but polarity instability limits long-term studies	Highly suitable for repeated-dose MNP exposure experiments
Throughput	Low	Moderate	High
Cost & Technical Demand	High cost, technically challenging	Moderate cost, medium technical expertise required	Lower cost, simpler culture methods
Quality Control	Requires immunostaining, RNA-seq, and single-cell QC	Requires polarity markers and barrier integrity tests	TEER, staining, and transcriptomic QC feasible

## 6. Leveraging Colon Organoid Models to Study MNP-Induced EOCRC

Human-derived colorectal organoids serve as a robust and physiologically relevant model for unraveling the potential mechanistic ties between MNP exposure, telomere dynamics, and colorectal carcinogenesis. Unlike two-dimensional culture systems, these three-dimensional structures preserve inherent patient-specific genetic and epigenetic profiles, making them ideal for investigating personalized responses to environmental insults. Organoid models enable controlled, chronic exposure to environmentally realistic concentrations of MNPs, allowing researchers to meticulously assess outcomes such as telomere shortening, DNA damage, oxidative stress, and cellular senescence. Such phenotypes can be quantified using techniques such as qPCR, quantitative FISH (Q-FISH), or single-telomere length analysis, while biomarkers including p16, p21, γ-H2AX, β-galactosidase activity, and chromosomal aberrations provide insight into senescence and transformation processes. Studies have already demonstrated the value of organoid systems in testing MNP toxicity and capturing inflammatory responses and proliferation shifts in colon models [86,91] (Table 3).

Experimental studies show that small PS particles (≈50–100 nm) penetrate and accumulate in human colon organoids, reduce epithelial viability, and induce transcriptional programs consistent with cytokine production and apoptosis (e.g., upregulation of *IL-6*, *IL-8*, *TNF-α*), supporting an organoid-based model of MNP-driven inflammation [86]. Distinct intracellular accumulation of ~50 nm polystyrene NPs in human intestinal organoids, with preferential uptake into secretory cell types, occurs via endocytic mechanisms and is associated with epithelial apoptosis and cytokine release, supporting a model in which particle uptake precedes epithelial injury [80]. In vivo and single-cell studies further show that chronic NPs exposure reshapes the colonic immune microenvironment (e.g., lysosome damage in macrophages, induction of IL-1β-producing macrophage subsets and skewing of Treg/Th17 responses), creating conditions that may favor tumor initiation and progression [92]. MNPs also induce oxidative stress and genotoxicity in intestinal models. Food-grade particulate additives and engineered nanoparticles (for example, TiO_2_/E171) provoke dose-dependent ROS production and DNA-strand breaks in small intestine and colon models and organoid-derived epithelia, as measured by comet assays and γH2AX staining; these insults are accompanied by transcriptional signatures implicating DNA-repair and redox pathways [93]. Moreover, studies of aminated polystyrene (PS-NH_2_) and other surface-modified NPs report ROS, mitochondrial damage and direct DNA cleavage activity in in vitro systems, indicating that particle chemistry, charge and size strongly modulate genotoxic potential [94]. Although direct demonstrations of MNP-induced cellular senescence in colon organoids remain sparse, intestinal and other organoid systems reliably model aging- and stress-associated senescence when maintained long-term or derived from aged donors. Long-term cultured colorectal and small intestine organoids and organoids derived from aged mice show growth defects, increased cell-cycle arrest (TGF-β/Smad3 → p16^INK4a signaling), and elevated senescence markers such as SA-β-gal, p21 and p16, supporting the use of these systems to study senescence induction and SASP emergence after environmental stressors [95]. Organoids are also an excellent substrate for epigenomic interrogation. Colon organoids derived from familial adenomatous polyposis (FAP) patients already demonstrate widespread differentially methylated regions and altered chromatin accessibility that mirror early CRC epigenetic signatures, indicating that organoids faithfully retain disease-relevant epigenetic states and are suitable for assessing environmentally triggered epigenetic remodeling [96]. While direct evidence of MNP-driven DNA methylation or histone-mark changes in colon organoids is still emerging, the platform readily supports ATAC-seq, ChIP-seq, bisulfite sequencing and integrated transcriptomics to detect such changes and to connect them to altered stem-cell programs or differentiation trajectories [96,97,98,99].

Finally, combination technologies substantially increase physiological relevance: microfluidic intestine-on-chip devices permit chronic, flow-based luminal MNP exposure and co-culture with immune or stromal cells, and CRISPR/Cas9-edited organoids allow testing of how EOCRC-relevant mutations (e.g., *APC*, *TP53*, *MLH1*) modify susceptibility to MNP-driven inflammation, genotoxic stress, senescence and epigenetic reprogramming. Together, these tools position colon organoids as flexible, patient-relevant platforms to test the mechanistic plausibility that environmental MNP exposure contributes to EOCRC initiation and progression [100].

Recent innovations in micro-organospheres (*MOSs*) highlight how next-generation culture systems can enhance organoid-based studies of environmental carcinogenesis. While existing MOS studies have primarily focused on oncology and drug screening, their technical features make them highly relevant for modeling chronic, low-dose exposures such as dietary MNPs. Wang et al. developed MOSs as miniature epithelial units encapsulated in hydrogel droplets, enabling rapid formation, high-throughput handling, and quantitative imaging while retaining essential tissue morphology and differentiation markers [101]. Deng et al. further demonstrated that patient-derived MOSs preserves inter-individual variation in drug sensitivity and maintain tumor–immune microenvironments, underscoring their ability to capture clinically relevant heterogeneity. Although these studies did not directly investigate MNPs, the scalability, throughput, and immune-competence of MOSs position them as an ideal platform for dissecting how MNP exposure interfaces with epithelial injury, immune dysregulation, and genetic susceptibility EOCRC [102]. Recent advances further include bioengineered human colon organoid models that integrate organoids with organ-on-a-chip microfluidic systems to generate “mini-colons” with in vivo-like architecture, enhanced cell type diversity (e.g., abundant mucus-producing goblet cells and mature colonocyte populations), and physiologically relevant luminal access, offering a valuable platform for more sophisticated mechanistic studies of colorectal epithelial responses to environmental exposures and disease processes [87]. Functional intestinal monolayers derived from organoids provide physiologically relevant epithelial barrier models suitable for mechanistic studies of environmental exposures [103]. Additionally, colon assembloids co-culture epithelial organoids with stromal cell types to recreate mature crypt architecture and epithelial–stromal signaling dynamics, offering enhanced physiological relevance for studies of gut homeostasis, inflammation, and tumorigenesis [104].

**Table 3 cells-15-00040-t003:** Colorectal/Small Intestine Organoid and Related Experimental Models Investigating MNP-Induced or Mechanistically Relevant Cellular Processes in EOCRC.

Process	Mechanistic Focus	Model Type/Assays	Key Findings and References
Inflammation	Cytokine release, immune activation	Human colon organoids exposed to MPs; apoptosis and immune gene expression	MNPs (50–100 nm) ↓ viability, ↑ inflammatory/apoptotic genes [86]
	Endocytic uptake and NF-κB activation	Intestinal organoids with PS-NPs; inflammatory response via endocytosis	PS NPs (~50 nm) accumulate, cause apoptosis/inflammation via clathrin-mediated endocytosis [80]
	Immune modulation and microenvironmental remodeling	Mouse colon microenvironment altered by NPs	NPs trigger IL-1β macrophages, Treg/Th17 skewing [92]
DNA Damage	ROS generation, strand breaks	iPSC-derived intestinal/colon organoids exposed to food-grade TiO_2_ (E171)	Dose-dependent ROS and DNA damage; altered DNA repair pathways [93]
	DNA cleavage and oxidative stress	Cell-free and epithelial cell assays with PS-NH_2_ NPs	PS-NH_2_ NPs cause DNA cleavage; size/surface chemistry matters [105]
Cellular Senescence	Stem cell aging, SASP	Long-term culture of intestinal/colon organoids and organoids from aged mice	Organoid systems allow potential study of senescence markers (p16, SA-β-gal), highlights the value of organoid models for studying aging [97,106]
Epigenetic Alteration	DNA methylation reprogramming in hereditary CRC risk	FAP-patient–derived normal colon organoids	Identified distinct methylation landscapes in FAP organoids, revealing early epigenetic changes linked to CRC predisposition [96]
	Multi-omics integration for MNP exposure profiling	Proposed organoid-based epigenetic platforms (ATAC-seq, ChIP-seq, RNA-seq)	Platform enables ATAC-seq/ChIP-seq/RNA-seq for epigenetic profiling [98]
Combined Stress Effects	Signaling modulation post MNP + radiation	Murine intestinal organoids with chronic low-dose NP exposure + radiation	Chronic NP exposure enhances TGF-β1/Smad signaling and amplifies inflammatory injury [107]

## 7. Conclusions

Collectively, accumulating evidence suggests that MNP exposure may represent an underrecognized environmental factor contributing to the rising incidence of EOCRC. By integrating mechanistic data from both experimental and organoid-based studies, it becomes clear that MNPs can disrupt colorectal epithelial integrity, promote oxidative and inflammatory signaling, and interfere with genomic and epigenomic stability—cellular stress responses that plausibly converge on EOCRC-relevant molecular pathways, including TP53 dysfunction, activation of stem-like and proliferative transcriptional programs, and APC-independent WNT signaling, which are central to early tumor initiation. The convergence of these cellular stress responses with host-specific factors, such as genetic susceptibility, microbiome composition, and dietary patterns, likely amplifies carcinogenic risk in younger populations. Thus, linking MNP-induced molecular perturbations with the distinct biology of EOCRC may reveal new biomarkers of early disease and novel avenues for prevention (Figure 4).

Looking forward, advanced organoid technologies provide an unprecedented opportunity to bridge environmental exposures with human pathophysiology. Although direct organoid-based studies examining MNP exposure in EOCRC are currently lacking, patient-derived and genetically stratified colorectal organoids are uniquely suited to mechanistically test whether MNP-induced oxidative stress, inflammation, and barrier dysfunction directly modulate *TP53* signaling, WNT pathway activity, and epithelial stemness in EOCRC-relevant contexts. The integration of patient-derived organoids with high-content imaging, microfluidics, and multi-omics profiling will allow real-time mapping of cellular responses to MNPs in a controlled yet physiologically relevant context. Such approaches not only refine our understanding of EOCRC etiology but also create a translational platform to screen environmental hazards and test therapeutic or dietary interventions. Ultimately, leveraging organoid-based systems to dissect the complex interplay between environmental plastic exposure and colorectal epithelial biology holds promise for advancing precision prevention strategies against EOCRC.

## 8. Challenges and Future Directions

Despite growing recognition of EOCRC as an urgent public health issue, several challenges hinder progress in prevention, diagnosis, and mechanistic research. Clinically, the absence of routine screening in individuals under 50 delays detection, while symptoms in younger patients are often misattributed to benign conditions, contributing to late-stage diagnoses. Research is further complicated by disease heterogeneity: EOCRC may exhibit molecular features distinct from late-onset CRC and even between its own subgroups, complicating biomarker discovery. Environmental risk factors, including diet, lifestyle, and emerging contaminants such as MNPs, remain difficult to evaluate due to limited longitudinal data and challenges in quantifying cumulative exposure.

Colorectal organoids provide a highly relevant platform to study EOCRC and environmental exposures; however, several barriers remain in their application to MNP research. These include: (i) lack of standardized exposure protocols, as variability in particle size, surface chemistry, and dosing complicates cross-study comparisons; (ii) incomplete incorporation of immune and stromal components, since most models currently lack immune cells, fibroblasts, and microbiota that are crucial for recapitulating in vivo responses; and (iii) limited longitudinal modeling, with chronic, low-dose exposure scenarios—more reflective of real-life MNP ingestion—still underexplored.

Recent advances in micro-organosphere (MOS) technology offer promising ways to address some of these limitations. Wang et al. developed MOSs as scalable, miniaturized 3D cultures that preserve tissue architecture while enabling rapid formation, high-throughput screening, and quantitative imaging [101]. Deng et al. further demonstrated that patient-derived MOSs capture inter-individual variability and maintain tumor–immune interactions, highlighting their potential for modeling complex disease contexts [102]. Although these studies did not investigate MNPs directly, the scalability, immune-competence, and throughput of MOSs make them well-suited for exploring chronic, low-dose environmental exposures such as MNPs, particularly in the context of EOCRC. A recent study developed a novel platform combining adult stem cell–derived small intestinal organoids with live fluorescence lifetime imaging microscopy (FLIM) to investigate how pristine MNPs interact with gut epithelium in real time. By generating “apical-out” organoids from porcine and mouse tissue, the authors enabled direct luminal exposure and demonstrated that <200 nm PMMA- and PS-based nanoparticles interact with apical and basal membranes in distinct, species-specific patterns. FLIM phasor analysis provided superior sensitivity over conventional fluorescence imaging, allowing “lifetime barcoding” to differentiate MNP types and their precise interaction sites within the organoid epithelium. Functional assays revealed that even pristine MNPs could alter mitochondrial membrane potential, affect total cellular energy budgets, and disrupt chemokine (CXCL-8) production after both short-term (24 h) and long-term (72 h) exposures. This work establishes apical-out organoids combined with FLIM as a powerful, physiologically relevant model for dissecting MNP uptake mechanisms and early epithelial stress responses—critical for understanding gastrointestinal exposure pathways relevant to conditions such as early-onset colorectal cancer [77].

Addressing these challenges will ultimately require integrative strategies that combine advanced organoid platforms with microfluidic “gut-on-chip” systems, immune and stromal co-cultures, and patient-derived models to more faithfully replicate human colorectal physiology. Such multidimensional approaches promise to uncover mechanistic links between environmental exposures and EOCRC risk, enabling biomarker discovery and the development of preventive interventions. Such systems may have to facilitate high throughput studies due to the vast variety of environmentally important MNPs, as gut and tissue interactions are likely to vary based on polymer type, morphology, and surface charge. Critical choice of the selected MNPs from a reference library mimicking environmental and exposure routes will have to be considered as well.

## Figures and Tables

**Figure 1 cells-15-00040-f001:**
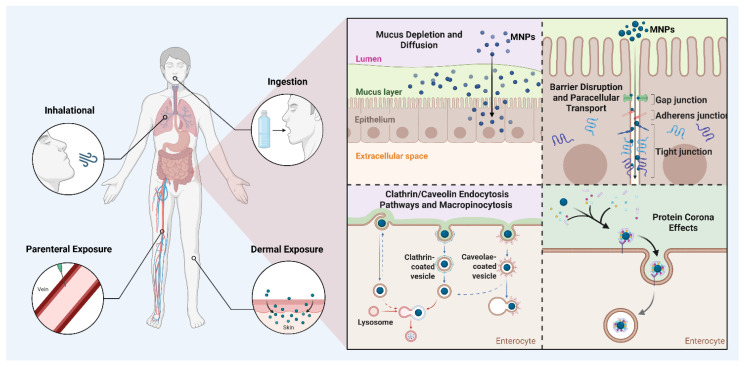
Overview of micro- and nanoplastic (MNP) exposure routes and uptake mechanisms. MNPs reach internal/colon tissues primarily through ingestion, inhalation, dermal, or parenteral routes. Within epithelial barriers, MNPs can cross through mucus disruption, tight junction alterations, or active endocytosis pathways (clathrin/caveolin-mediated and macropinocytosis). Protein corona formation further influences their cellular interaction, uptake efficiency, and potential toxicity. Created with BioRender.com.

**Figure 2 cells-15-00040-f002:**
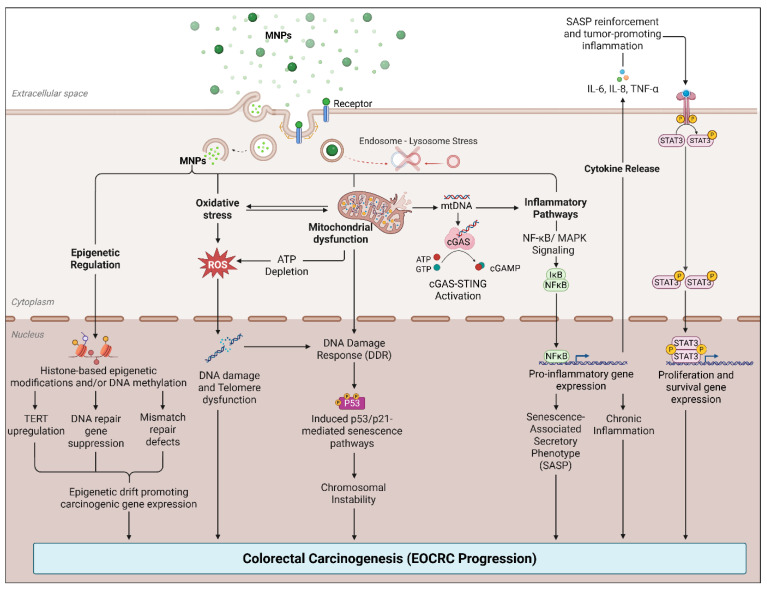
Proposed model of MNP-induced chromosomal instability and carcinogenesis in the colon. Created with BioRender.com.

**Figure 3 cells-15-00040-f003:**
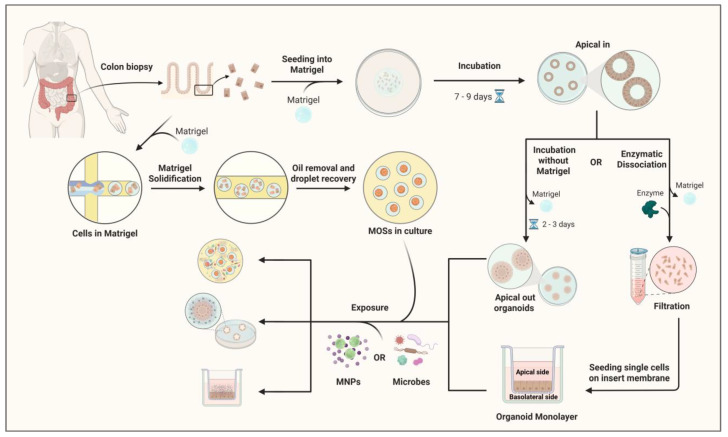
Schematic overview of the workflow for exposing human colon apical-out organoids and monolayer cultures to MNPs. Human colon biopsy samples establish 3D colon organoids embedded in Matrigel and cultured for 7–9 days. Organoids either remain in Matrigel (apical-in configuration) or undergo two downstream approaches: in the first, removal of Matrigel and incubation for 2–3 days generate apical-out organoids; in the second, enzymatic dissociation into single cells, filtration through a cell strainer, and seeding onto transwell inserts form organoid-derived monolayers. In parallel, colon epithelial cells can also be encapsulated in Matrigel droplets to generate microorganospheres (MOSs), which maintain epithelial polarity and allow co-culture with microbiota under controlled conditions. MOSs are subsequently recovered and can be used for exposure experiments alongside apical-out and monolayer models. Both cultures are exposed to MNPs. Organoids and conditioned media are collected for downstream analyses, including gene expression profiling, telomere length assay, immunostaining, proteomics, and metabolomics. Created with BioRender.com.

**Figure 4 cells-15-00040-f004:**
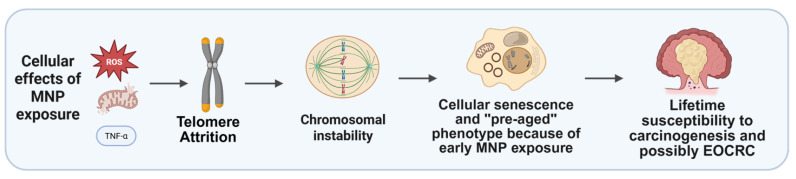
Hypothetical cascade of MNP effects. After MNP exposure telomere-related changes could drive chromosomal instability and cellular senescence, possibly predisposing to early-onset colorectal cancer. Created with BioRender.com.

**Table 1 cells-15-00040-t001:** Somatic Molecular and Biological Features in EOCRC and AOCRC [3,49,50,51].

Feature/Pathway	EOCRC	AOCRC
Microsatellite instability (MSI)	MSI in EOCRC (~10–15%), mostly hereditary (Lynch syndrome); sporadic MSI rare	MSI in AOCRC (~20%), mainly sporadic via MLH1 methylation.
CpG Island Methylator Phenotype (CIMP)	CIMP-high less prominent in EOCRC; when present, often in right-sided or BRAF-mutant subset	Common in older patients, often linked with BRAF mutations and serrated pathway
BRAF (V600E) mutation	Rare (<5%)	Common (~10–20%)
*KRAS* mutations	Moderate (30–40%)	Similar or slightly higher frequency (~40–50%). Note: While AOCRC may have a slightly higher frequency, *KRAS* remains a major player in both EOCRC and AOCRC.
*TP53* mutations	More frequent (~60–70%); often early event	Common (~50–60%)
*APC* mutations	Less frequent and occur later; WNT activation via alternative mechanisms (e.g., RNF43 loss)	Very frequent (~80%) early driver event
*PIK3CA* mutations	Less common	More common (~15–20%)
Chromosomal instability (CIN)	Present but via different routes; more focal copy number changes	Classic CIN pathway with extensive aneuploidy
Epigenetic alterations	Distinct methylation patterns; lower age-related methylation drift. Note: Research has identified unique epigenetic alterations in EOCRC, sometimes linked to specific patient populations or racial/ethnic backgrounds.	Strong age-related methylation changes (epigenetic drift)
Tumor location	Predominantly left-sided (rectum, rectosigmoid, sigmoid colon)	More evenly distributed; more right-sided tumors with increasing age
Immune microenvironment	Enhanced immune infiltration and inflammatory signatures even in MSS tumors. Note: New research explores differences in T-cell receptor diversity, with higher diversity observed in EOCRC, suggesting distinct immune responses based on age of onset.	Variable; immune infiltration higher in MSI-H tumors only
Senescence and aging markers	Altered senescence pathways; upregulation of senescence associated secretory phenotype (SASP)-related genes	Accumulation of aging related senescence gene products.
Telomere dynamics	Shorter tumor telomeres associated with telomerase activation and chromosomal instability, longer tumor telomeres. Alternative lengthening of telomeres (ALT) and chromosomal stability (CSS) of tumor DNA	Shorter tumor telomeres, telomerase activation and chromosomally unstable tumor DNA; Longer tumor telomeres, ALT and CSS
Histology	Higher proportion of poorly differentiated tumors, mucinous carcinomas, and signet-ring cell carcinomas, which indicates more aggressive tumor biology.	Lower prevalence of aggressive histological features like mucinous and signet-ring cell carcinomas.
Gut microbiome association	Increased prevalence of Fusobacterium nucleatum, Bacteroides fragilis, Peptostreptococcus anaerobius	Microbiome dysbiosis, but less distinct composition. Note: Research continues to show distinct microbial profiles between EOCRC and AOCRC, though with some inconsistency across studies. Geographic location and diet are recognized as important confounding factors.
Pathways implicated	DNA damage response, metabolic reprogramming, immune regulation	Classic adenoma–carcinoma sequence via APC–KRAS–TP53 axis

## Data Availability

No new data were created or analyzed in this study.

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
