# Peer review of "Colon Organoids as Experimental Models to Study the Effect of Micro-Nanoparticles as a Driver of Early-Onset Colon Cancer"

_cells, 2025, doi:10.3390/cells15010040_

Round 1
Reviewer 1 Report
Comments and Suggestions for Authors
REVIEWER COMMENTS
In the current work, Heydari Z. et al., present a comprehensive review that aims to summarize two rapidly evolving areas: the emergence of early-onset colorectal cancer (EOCRC), and the biological impact of exposure to micro- and nanoplastics (MNPs). The figures are clear and well designed, and the authors have compiled an extensive amount of data from the literature. Although the topics are of potential biological relevance, the manuscript reads more like two separate reviews, or even three if including the use of small intestinal and colonic organoids. The central connection between the above-mentioned topics is only partially developed.
Specific Comments
- A critical aspect of the review is the limited mechanistic integration between MNPs and EOCRC. Although ROS production, stress, epithelial barrier dysfunction, etc. are well described, these pathways are poorly linked to EOCRC-specific molecular features described in the text, such as higher TP53 mutational burden, stem-like and proliferative transcriptional programs or APC-independent WNT activation. The manuscript would be improved by discussing how MNPs-induced processes, either in in vivo or in vitro using organoids tools, could be linked to these EOCRC-relevant pathways.
- Several paragraphs in the text imply a direct relation between MNPs exposure and EOCRC. However, the references cited by the authors do not directly support that relation and suggest just plausibility/possibility; thus, those sections in the manuscript should be moderated accordingly.
- Lines 98-99,104-107, and 120-122 repeat the same concepts, and words. This issue should be corrected.
- The term PS-MPs is used without definition; all abbreviations should be clearly defined.
- One referenced study (Ref.58) reports both loss of goblet cells and increased mucus secretion. This contradiction should be briefly discussed by the authors.
- The organoid section would be improved by including more sophisticated and physiologically relevant systems, including a recently developed tool by Mitrofanova O. et al., Cell Stem Cell, 2024. This model more closely mimics the native architecture of the tissue and is highly relevant for the kind of study proposed by the authors.
- Please, clarify whether apical-out culture without Matrigel is broadly feasible in normal organoids or whether this method is only supported in CRC-derived models.
- The telomere biology section is fine but feels disconnected from the rest of the manuscript. It could be better reorganized into clearer subsections including an explicit link with MNPs and EOCRC molecular pathways.
- Standardize the terminology used for colonoids vs. colorectal organoids or human colonic organoids.
- Please, ensure a clear distinction when mentioning small intestine or colon organoids. It is important, because small intestinal cancers are not as frequent as colon cancers. On the contrary, small intestine organoids could be used as models for regeneration- or inflammatory-related diseases (i.e. Crohn’s disease).
Reviewer 2 Report
Comments and Suggestions for Authors
In this comprehensive review, Heydari et al discuss the use of colonic organoids as a model to explore how MNPs impact the pathophysiology of EOCRC. They discuss routes of MNP exposure, how they are internalized by intestinal epithelial cells and underlying mechanisms to explain their toxicity. The review provides a nice summary of known genetic differences between EOCRC and AOCRC and the potential role of telomeres in early-onset disease. The authors discuss the major organoid models, strengths and limitations of each, and how they may be used to study MNP-induced EOCRC. The authors nicely summarize challenges of using organoid systems and potential future directions. Overall, the review was thorough, well written and will serve as a valuable resource to the EOCRC community. Below are a few recommendations that should be considered to improve readability and ease of readers to find source data.
- The two organoid-based studies (lines 163 and lines 169) should be moved to later in the review after organoids are introduced and are more thoroughly discussed.
- Consider swapping sections 5 and 6 in terms of their order in the review. While section 6 on telomeres is important, placing section 5 (intro to organoids) immediately before section 7 (how to use them to study MNPs) would improve overall flow.
- References should be added to Tables 1 and 2 to direct the readers to the relevant literature used to support the data presented. These can be review articles.
- Lines 38, 40, 75, 132, 142, 247 need references. Check appropriateness of ref 34 (miRNA carrier) for line 148 and 53 for line 250.
- Lines 98 and 99 are redundant, combine.
- Lines 104 and 106 are redundant, combine.
- Lines 112 and 113 should be combined. Without reference to MPs, as line 112 currently reads, it is not clear what NPs are being compared to.
- Line 131: Size specific intestinal effects “of MNPs”
- Line 147: … reported to “have” higher uptake
- Line 382: effects “that were” more pronounced.
- I didn’t see that Table 3 was referred to in the text, but I might have missed it.
Round 2
Reviewer 1 Report
Comments and Suggestions for Authors
The responses provided address the points raised in the initial review, and the revisions improve clarity and organization of the manuscript. No further comments are raised